# Effects of High-Intensity Laser Therapy (HILT) on Skin Surface Temperature and Vein Diameter in Healthy Racehorses with Clipped and Non-Clipped Coat

**DOI:** 10.3390/ani13020216

**Published:** 2023-01-06

**Authors:** Paulina Zielińska, Maria Soroko-Dubrovina, Karolina Śniegucka, Krzysztof Dudek, Nina Čebulj-Kadunc

**Affiliations:** 1Department of Surgery, Faculty of Veterinary Medicine, Wroclaw University of Environmental and Life Sciences, Plac Grunwaldzki 51, 50-366 Wrocław, Poland; 2Institute of Animal Breeding, Wroclaw University of Environmental and Life Sciences, Chelmonskiego 38C, 51-630 Wrocław, Poland; 3Center for Statistical Analysis, Wroclaw Medical University, Marcinkowskiego 2-6, 50-368 Wrocław, Poland; 4Institute of Preclinical Sciences, Veterinary Faculty, University of Ljubljana, Gerbičeva 60, 1000 Ljubljana, Slovenia

**Keywords:** high-intensity laser therapy, carpal joint, physiotherapy, thermography, clipped coat, skin temperature

## Abstract

**Simple Summary:**

The use of high-intensity laser therapy (HILT) has recently been introduced in equine veterinary medicine. The goal of the research was to evaluate the effects of HILT on skin surface temperature and vein diameter in the carpal joint region in 20 racehorses divided into two groups: with clipped (*n* = 10) and non-clipped coat (*n* = 10) in the area of application. The hypothesis was that HILT would lead to a greater increase in skin surface temperature and vein diameter in the clipped coat group than in the non-clipped group. Variations in vein diameter and skin surface temperature were assessed before and after HILT. The HILT treatment resulted in a greater increase in skin surface temperature in the non-clipped coat group horses compared to the clipped group, while vein diameter increased more in the clipped group horses compared to the non-clipped group. Larger increases in skin surface temperature can be achieved with a non-clipped coat. Coat clipping of the treatment area increases vein diameter, while reducing skin surface temperature. Further research is needed to specify the parameters for treatment of skin areas with clipped and non-clipped coat in order to perform effective laser therapy.

**Abstract:**

The aim of this study was to investigate the differences in the effects of high-intensity laser therapy (HILT) on skin surface temperature and vein diameter in the carpal joint region in racehorses with clipped and non-clipped treatment areas. The study included 20 Thoroughbreds split into two equal groups: clipped coat and non-clipped coat. Horses underwent thermographic examination to detect changes in skin surface temperature at the medial surface of the carpal joint, followed by ultrasonographic examination to assess changes in the diameter of the medial palmar vein before and after HILT. The increase in skin surface temperature after HILT was significantly lower in the group with clipped coat than in the non-clipped group. The group with clipped coat showed a greater increase in vessel diameter. There was a significantly weak negative correlation between the changes in average skin surface temperature and vein diameter in both groups. In conclusion, an efficient photothermal effect can be achieved in skin with a non-clipped coat and clipping the treatment area increases photobiostimulation of the tissue, while reducing the photothermal effect. Further research is needed to specify the parameters for the treatment of skin with clipped and non-clipped coat in order to perform effective laser therapy.

## 1. Introduction

High-intensity laser therapy (HILT) is a treatment method that has gained popularity in equine veterinary medicine and physiotherapy in recent years [1,2]. The treatment consists of high peak power (1–3 kW) with a wavelength of 1064 nm, a low frequency (30 Hz) and an average power of 6 W with a very short pulse duration of less than 150 ms, which induces radiation transmission into deeper tissue layers, resulting in diffusion of the laser energy in the irradiated soft tissue [3,4].

HILT has a photothermal effect, which leads to an improvement in microcirculation, blood vessel permeability and metabolic process via vasodilatation [5]. Increased blood flow also promotes oxidative reactions, higher adenosine triphosphate production and the activity of destructive enzymes such as collagenase. It also influences catabolic rates, while accelerating tissue regeneration [6,7,8]. In addition, the generation of reactive oxygen species during laser therapy can stimulate the endogenous production of antioxidants, the differentiation of progenitor cells and cell growth [9].

Previous studies have also reported that HILT has biostimulatory, anti-inflammatory and analgesic properties, without adverse effects on or histological risks to tissue. Photomodulation of HILT depends on the effects of photons on tissue, absorbed by cytochrome c oxidase in mitochondria. Cellular dissociation of nitric oxide from cytochromes increases metabolic turnover and vasodilatation [10]. High-power laser photobiomodulation of mesenchymal stem cells has been found to have an anti-inflammatory effect, enhancing the therapeutic properties of such cell treatment without compromising cellular survival [11]. Biostimulatory phenomena occur gradually as a part of local circulatory effects, leading to faster tissue repair [12] and achievement of the analgesic effect [13].

There are an increasing number of randomized controlled trials reported in the literature based on clinical use of HILT in equine veterinary medicine. In vivo experiments have demonstrated the biological effects of HILT in the treatment of clinical cases, including tendinopathy and desmopathy [14,15,16] and osteoarthritis [17]. Other studies have focused on the application of HILT in the treatment of carpal medial collateral ligament [18] and back pain [19].

In the future, research needs to be expanded to evaluate the effectiveness of HILT on healthy tissue to specify the precise parameters for injury therapy more accurately. The question of which parameters of laser therapy are most effective in achieving optimal tissue improvement is currently not clear. Effective laser treatment requires a number of parameters, including power and single irradiation time, which contribute to the total emitted and absorbed photon dose [20]. Based on human studies it was found that the optical windows for the wavelength of laser light absorbed by cytochrome c oxidase have emissions between 600 and 1000 nm. Photons with a wavelength of less than 600 nm are strongly absorbed by the major chromophores (melanin and hemoglobin), while photons with a wavelength of more than 1000 nm are absorbed by water, causing tissue destruction [9]. In a previous study, we demonstrated that after HILT, the temperature of the pigmented skin surface increases, while the temperature of the non-pigmented skin surface decreases. In addition, the vein diameter was found to increase after HILT in horses with both pigmented and non-pigmented skin, although the variations between the groups was not significant [21]. It is therefore clear that the amount of melanin in the epidermis has a key role in the absorption of light energy. In a study by Duesterdieck-Zellmer et al. [22], the penetration of laser energy into the digital flexor tendons of horses was greater in non-pigmented than in pigmented skin. In a study on dogs, it was found that coat color and density can influence laser light penetration and absorption by the target tissue. According to the low-profile thermopile sensor measurements, clipping the coat significantly contributes to increased laser light transmission in the area of the lateral side of the inguinal fold and calcaneal tendon compared to non-clipped coat. The higher laser transmission in the clipped target tissue is presumably due to lower photon scattering, reflection, and hair absorption [10]. Similar results were found in a study of the properties of low-level laser therapy in horses, which reported that laser penetration was significantly reduced when transmitted through non-clipped coat in a tendon area [23].

However, none of the above studies addressed the differences between HILT on clipped and non-clipped treatment areas in horses. Accordingly, the objective of the present study was to evaluate the differences in the influence of HILT on skin surface temperature and vein diameter of the medial side of the carpal joint region in clinically healthy racehorses with pigmented clipped and non-clipped coat in the treatment areas. The hypothesis of the study was that HILT results in a greater increase in skin surface temperature and vein diameter in the clipped coat group, compared to the non-clipped coat group. The medial side of the carpal joint was selected for anatomical evaluation to efficiently examine the diameter of the superficial veins.

## 2. Materials and Methods

Prior to the start of the study, the project protocol was approved by the Local Ethics Committee for Animal Experiments in Wroclaw, Poland (No. 003/2020).

### 2.1. Animals and Study Design

The study was conducted on 20 Thoroughbreds aged 3 to 4 years in August 2021. All horses were trained in the season for flat racing at Partynice Racecourse (Poland) and were housed in individual boxes in one stable with the same management and training schedule. All horses were healthy at the clinical examination prior to the study. None of the horses had an injury to the right forelimb, where HILT treatment was applied. The diagnosis of possible pathological disorders was based on interviews with the trainers and visual and manual assessments performed by a veterinarian (P.Z.).

The horses selected for that study were black, bay, and chestnut, without any white marks on the right front limb (close to and in the treatment area). Skin pigmentation in the treatment area was assessed as pigmented. The horses were divided into two groups: clipped coat (*n* = 10) and non-clipped coat (*n* = 10). On the day of the experiment, an electric clipper (blade size 0.8 mm) was used to shave the treatment area in the clipped group. A 10 cm^2^ area of skin was shaved for all horses.

The horses of both groups were subjected to an examination procedure based on previous studies [21,24]. On the day of the examination, each horse was first subjected to a thermographic and then to an ultrasonographic recording. Both procedures were performed shortly before and immediately after HILT to detect changes in skin surface temperature at the medial surface of the carpal joint and the diameter of the medial palmar vein. Horses were examined at rest and before any exercise under saddle or in the walker. All measurements were taken with the horses standing still in the stable corridor with weight bearing right front limbs.

### 2.2. High-Intensity Laser Therapy

HILT was applied using a class IV Polaris HPS laser (Astar, Bielsko-Biała, Poland), which delivers two wavelengths simultaneously: 808 nm (an AlGaAs laser with a maximum output power of 8 W) and 980 nm (an InGaAs/AlGaAs laser with a maximum output power of 10 W). To avoid uncontrolled thermal effects, tissue destruction and skin burns, various treatment parameters were utilized for each wavelength. The energy density was 30 J/cm^2^ at a power of 4 W and a pulse mode frequency of 1000 Hz. The treatment area was localized on the medial aspect of the right carpal joint at the level of the accessory carpal bone, the area was 10 cm^2^, and total irradiation time was 93 s. HILT was applied by manual scanning of the treatment area with the probe placed perpendicular to the treated area in firm contact with the skin. All treatments were performed by the same investigator (P.Z.). To avoid scattering of the laser light, a contact technique was used without applying pressure to the tissue and with the least possible amount of contact of the probe on the horses’ coats or skin.

### 2.3. Ultrasonographic Examination

Ultrasonographic examination was performed over the medial palmar vein using a 10 MHz linear transducer (Dramiński^®^, Olsztyn, Poland), with a minimal amount of coupling gel and pressure on the tissue to avoid false changes in vessel diameter. Forty ultrasound scans were blindly evaluated by the same person, measuring the vein diameter in the transverse view at the level of the distal half of the accessory carpal bone, where the vessel is located most superficially (Figure 1). The on-board calipers of the Dramiński Blue Ultrasound Scanner (SVN_WCREV = 10462, Dramiński^®^, Olsztyn, Poland) could measure with an accuracy of 0.1 cm. All measurements were carried out at the same time, three times, and the average value was calculated.

### 2.4. Thermographic Examination

The thermographic examination was performed with a VarioCam HR infrared camera (uncooled microbolometer focal plane array; resolution, 640 × 480 pixels; spectral range, 7.5–14 mm; InfraTec, Dresden, Germany). The protocol for the examination was the same as previously described by Zielińska et al. [21] and Soroko et al. [25]. To minimize environmental influences, the examinations were always performed in an enclosed stable. Horses were acclimated to the enclosed stable for 20 min. The distance of the animal from the camera was set at 1 m for all recordings, with the emissivity set at 1 for all measurements [26]. The average surface temperature of the rectangular area over the investigated region (Figure 2) was determined using the software IRBIS 3 Professional (InfraTec, Dresden, Germany), as previously described [21].

### 2.5. Statistical Analysis

The Shapiro–Wilk test was used to assess the compliance of the empirical distributions of skin temperature and vessel diameter with the theoretical normal distribution. Since in some groups these distributions deviated from the normal distribution, these parameters were given as medians (Me) and the lower (Q1) and upper (Q3) quartiles.

The hypotheses about the absence of a difference between the average skin temperature values and the blood vessel diameters in the two groups of horses were verified with the Mann–Whitney U test. To assess the strength, direction, and significance of the relationship between changes in skin surface temperature and vein diameter, the value of Spearman’s rank correlation coefficient (rho) was estimated. In addition to the significance level *p*, the power of the test 1 – β was also estimated (Table 1). The satisfactory values were *p* < 0.05 and 1 − β > 0.8. STATISTICA (version 13.3, TIBCO Software Inc., Palo Alto, CA, USA) was used for the statistical analysis.

## 3. Results

All horses examined completed their HILT treatment and measurements. There was a statistically significant difference between the clipped and non-clipped group in terms of skin surface temperature of the medial side of the carpal joint before HILT application. No statistically significant differences were found between the groups with regard to the diameter of the medial palmar vein before applying HILT. After the application of HILT, there were no statistically significant differences in skin surface temperature or vein diameter between the clipped and non-clipped groups (*p* = 0.089 and *p* = 0.130, respectively) (Table 1).

A comparison between the clipped and non-clipped groups, in terms of changes in skin surface temperature and vein diameter before and after HILT, showed that after HILT the increase in skin surface temperature was significantly lower in the clipped coat group than in the non-clipped coat group (2.1 °C versus 3.9 °C, *p* = 0.006). The clipped coat group had a greater increase in vessel diameter than the non-clipped coat group (0.8 mm vs. 0.1 mm, *p* < 0.001) (Figure 3).

A weak negative correlation (rho = −0.496) was also found between changes in the average skin surface temperature and vein diameter in both groups of horses.

## 4. Discussion

One of the biggest challenges in integrating HILT into established medical practice is providing consistent treatment parameters and patient preparation recommendations. There are few studies examining the physiological effects of HILT on different types of healthy tissue or procedure parameters such as wavelength, energy density, treatment duration, and individual patient characteristics [5,21,24].

Previous studies conducted on clinically healthy horses have shown significant HILT effects in increasing blood vessel diameter and skin surface temperature in the clipped treatment area of a clinically healthy tarsal joint [24] and have shown how to enhance the photothermal effects of HILT in healthy horses [5]. However, these studies did not compare the effects of HILT between clipped and non-clipped coat groups. Other studies comparing the effects of HILT on vein diameter and surface temperature in healthy racehorses with clipped coat found that HILT caused an increase in surface temperature of pigmented skin and a decrease in surface temperature of non-pigmented skin, with vein diameter increasing in both groups [21].

In the present work, the hypothesis that HILT leads to a larger increase in skin surface temperature in the group of horses with clipped coat could not be confirmed. However, it could be clearly shown that HILT leads to a significant increase in the skin surface temperature of the non-clipped coat group. The influence of shaving on increased laser transmission has been discussed in previous studies on low-level laser therapy. It is assumed that the lack of a hair coat contributes to a lower scattering of photons and that melanin reduces photon absorption [22,23]. According to Laakso et al. [27], laser light passing through a coat can cause reflection, absorption, and scattering of photons, reducing or preventing their delivery to the target tissue. Some authors have claimed that hair should be clipped before laser treatment, because it absorbs 50 to 99% of the light [28]. Similar to the results of the present study, Bergh et al. [29] found that laser treatment caused a higher increase in skin temperature in non-clipped fetlock joints than in clipped ones. It is possible that clipping allows for better photon absorption, but that the lack of a thermal layer leads to heat loss and thus contributes to a less efficient increase in skin temperature. It has been well documented in previous studies that the hair coat of horses absorbs infrared radiation emitted from the skin surface [30]. The air trapped in the hair coat is a poor conductor of heat and provides efficient insulation against excessive heat loss through the skin. Turner et al. [31] assessed the influence of clipping the coat on changes in body surface temperature and demonstrated that clipped areas are warmer than unclipped areas. Skin with clipped coat radiates more heat because it lacks a thermal layer.

In the present work, it was also found that the group of horses with clipped coat showed a greater increase in vascular diameter than the non-clipped group, which confirms the hypothesis established at the outset. Increases in vessel diameter depend on the photothermal mechanisms that take place in irradiated tissue. Nagasawa [32] has proposed that the laser beams act directly on the sympathetic nervous system, contributing to a decrease in the diameter of blood vessels, followed by vasodilation. The effect of the laser energy on the blood vessels is determined by their distance from the treatment area. It is possible that the coat absorbs the laser light preventing the proper functioning of the autonomic nervous system. All horses had pigmented (dark) skin in the treatment area and vasodilatation was observed in both study groups, however the increased vein diameter was significant only in the clipped coat group. Thus, the decision whether or not to clip a coat depends on the desired HILT effect. A biostimulatory HILT effect was found in previous studies where vein diameter increased following treatment in both horses with pigmented and non-pigmented skin [21].

A weak negative correlation was also found between changes in vein diameter and average skin surface temperature in both groups of horses. Opposite results were found in a previous study, which found a correlation between the increase in skin surface temperature and vein diameter after HILT at the tarsal joint [24]. In another study based on pigmented and non-pigmented skin, no correlation was found between changes in skin surface temperature and vein diameter in either group of horses [21].

The present study was limited by its small sample size. It included only 20 horses, which may have limited the statistical results. Another limitation might have resulted from the short-term HILT effects studied. In the present study, only the thermal and biostimulatory effects immediately after laser therapy were investigated, without evaluating the effects of HILT on the treated areas in the long term.

## 5. Conclusions

This study has shown a significant increase in skin surface temperature in a group of horses with non-clipped coat at the treatment area compared to a clipped coat group, with a smaller increase in vessel diameter. This suggests that the hair coat plays an important role in the transition and absorption of light energy. It can be concluded that a more efficient photothermal effect can be achieved on the skin with non-clipped coat and that clipping of the treatment area increases the photobiostimulation of the tissue while reducing the photothermal effect. Further research is needed to specify the parameters for the treatment of skin with clipped and non-clipped coat in order to perform effective and safe laser therapy.

## Figures and Tables

**Figure 1 animals-13-00216-f001:**
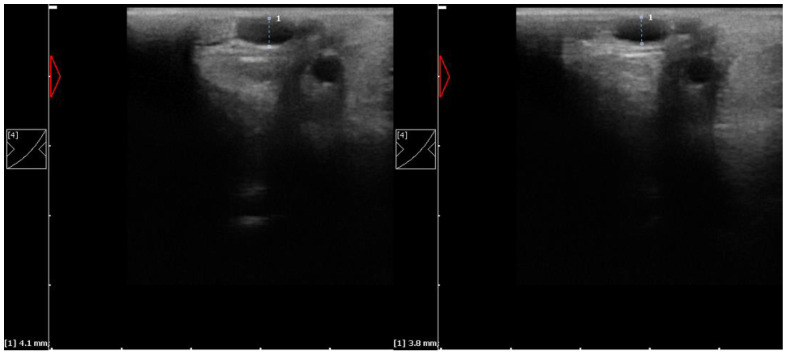
Transverse ultrasound images of the medial palmar vein, with diameter measurement of the right carpal joint taken before (**left**) and just after HILT (**right**).

**Figure 2 animals-13-00216-f002:**
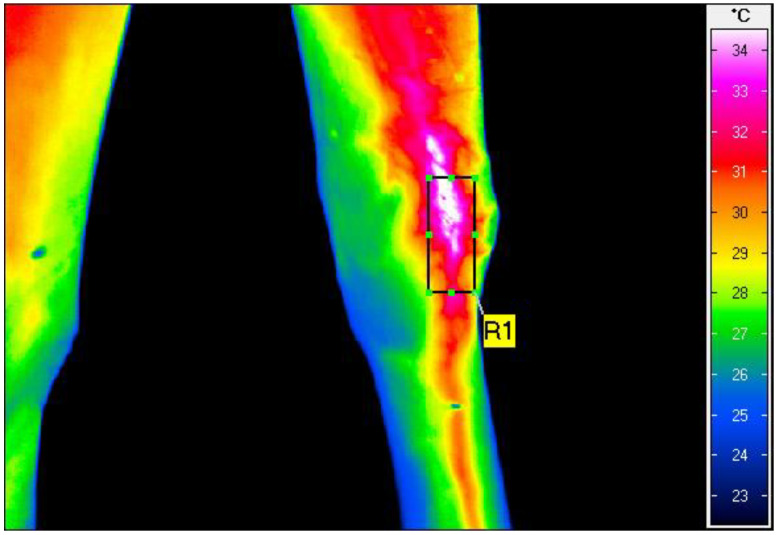
Thermographic image of the medial view of the right carpal joint, taken after HILT treatment in the non-clipped group. The rectangular area (R1) shows the average skin surface temperature as 31.9 °C.

**Figure 3 animals-13-00216-f003:**
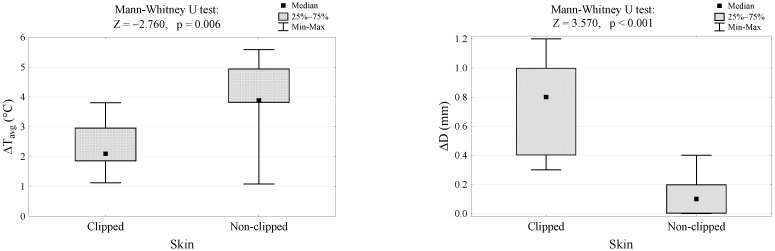
Differences in skin surface temperature and vein diameter after HILT in the clipped and non-clipped coat groups. ∆Tavg, differences between values of skin surface temperature after HILT; ∆*D*, differences between values of vein diameter after HILT.

**Table 1 animals-13-00216-t001:** Skin surface temperature, vein diameter and differences in skin temperature and vein diameter values (median (Q1–Q3) and range (Min–Max)) before and after high-intensity laser therapy (HILT) application to horses with clipped and non-clipped coats.

	Parameters	Clipped Coat (*N* = 10)	Non-Clipped Coat (*N* = 10)	Clipped Coat vs. Non-Clipped Coat (*p*-Value)	Power 1 − β
Before HILT	T_avg_ (°C)	31.6 (29.0–33.0)	27.9 (27.3–29.0)	0.005 **	0.974
	Range	28.9–33.4	28.9–33.4		
	*D* (mm)	4.6 (4.0–4.9)	4.6 (4.0–4.9)	0.939	0.051
	Range	3.6–5.2	3.5–5.3		
After HILT	T_avg_ (°C)	33.4 (32.5–34.9)	32.4 (31.3–33.5)	0.089	0.562
	Min–Max	31.1–35.4	29.5–33.8		
	*D* (mm)	5.3 (4.8–5.8)	4.6 (4.2–5.1)	0.130	0.796
	Range	4.0–6.4	3.5–5.6		
	∆T_avg_ (°C)	2.1 (1.9–3.0)	3.9 (3.8–5.0)	0.006 **	0.863
	Range	1.1–3.8	1.1–5.6		
	∆*D* (mm)	0.8 (0.4–1.0)	0.1 (0.0–0.2)	<0.001 ***	0.999
	Range	0.3–1.2	0.0–0.4		

** *p* < 0.01, *** *p* < 0.001 (Mann–Whitney U test); Abbreviations: HILT, high-intensity laser therapy; T_avg_, skin surface temperature; *D*, vein diameter; ∆T_avg_, differences between values of skin surface temperature after HILT; ∆*D*, differences between values of vein diameter after HILT.

## Data Availability

The data presented in this study are available on request from the corresponding author. The data are not publicly available for privacy reasons.

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
