# Peer review of "Effects of High-Intensity Laser Therapy (HILT) on Skin Surface Temperature and Vein Diameter in Healthy Racehorses with Clipped and Non-Clipped Coat"

_animals, 2023, doi:10.3390/ani13020216_

Round 1
Reviewer 1 Report
This is a nice simple study that is well described. It just needs some tidying, specifically:
Line | Comment |
23 | Change 'grater' to 'greater' |
25 | Please clarify what is meant by an 'efficient increase in skin temp' |
25 | Delete the second 'in skin' |
25 | Suggest insert 'a' between 'with non-clipped' |
25-26 | Suggest split this sentence into two, with one point per sentence |
31 | Suggest change 'area' to 'areas' |
39 | Suggest replace 'can be achieved with skin with non-clipped' with 'can be achieved in skin with a non-clipped' |
49 | The k in KW should be in lower case |
49, 50 | Suggest be consistent with spacing between numbers and units |
65 | cells been' should be 'cells has been' |
77 | The question which parameters' should be 'The question of which parameters' |
81 | Suggest delete comma |
83 | Keep the number and the unit together with a non-breaking space |
124, 139 | The 2 should be superscripted |
131 | with the horses stood still in stable corridor' should be 'with the horses standing still in the stable corridor' |
158 | Delete the first 'and' |
167 | Remove the underline from the degree symbol |
176-178 | Suggest rephrase to be 'In addition to the significance level p, the power of the test 1 - β was also 177 estimated (Table 1).' |
181-188 | Results could be put much more simply |
191 | Suggest rephrase to be 'therapy (HILT) application to horses with clipped and non-clipped coats.' |
196 | Suggest insert comma between 'HILT' and 'showed' |
198 | Remove the underline from the degree symbol |
198 | Compare the spacing between the number and the degree C symbol with Line 167 and make it consistent. This needs to be corrected throughout the manuscript. |
205 | Use consistent spacing either side of the symbols |
207 | Correct the degree symbol |
210 | Rho compared with rho in line 207; please be consistent |
210-211 | Suggest delete both commas, they are not needed here |
214 | Delete comma after practice |
215 | Delete 'of' |
231 | Insert 'the' between 'of' and 'non-clipped' |
232 | Delete comma |
261-266 | Do you have any suggestions why the results you received might be different from the other studies you mention? |
266 | Delete comma before bracket |
301 | Insert space between the year and the volume |
316 | Insert space between the volume and the pp |
324 | Should '247e54' be '247–254'? |
354 | Should 'Canince' be 'Canine'? |
356 | Format of CO2 |
357 | Change the hyphen to an en dash in the page range |
Reviewer 2 Report
This was a well-designed study. There is some missing information that should be included.
Lines 75 – 84: Most of this paragraph seems unnecessary in the introduction. I am not sure how this relates to skin temperature and vein diameter. Consider revising/deleting. It seems like the end of this paragraph is about patient factors. The beginning of the paragraph should reflect this.
Line 114: Please include some more information on the horses. Please include the distribution of horse coat and skin colors. What time of year was the study performed (i.e., what was the length of hair coat)?
Lines 134-143: Was the laser probe moved during treatment or was it held statically? What was the area of treatment? Please include more area about the treatment area location.
Lines 146-150: At what level was the vein diameter measured. Were still images stored? What software was used for measurements? Was a single measurement taken or multiples? Were all measurements done at the same time or in batches?
Line 154: Was thermography performed before ultrasound or after? They should be listed/described in the materials and methods in the order that they were performed.
Line 159: Was the horse acclimated to the enclosed stall for some set period of time?
Lines 169-172: If the data were normally distributed (Shapiro Wilk test), why were medians and lower/upper quartiles calculated? It seems that these should be means, 95% confidence intervals.
Table 1: Please check the values in this table (After HILT – D (mm) – the median value is 4.0 with Q1-Q3 (4.8-5.8). This does not seem correct.
Figure 3 is repetitive of what is presented in Table 1. Please delete this figure.
Lines 204-205: This sentence needs to be reworded. This test does not tell you about whether the sample size is too small. It tells you that there is not a strong relationship between these 2 characteristics.
Lines 205-207: This correlation coefficient value does not indicate a strong correlation. Please delete this statement as well as the following one about the increase in temp and decrease in vein diameter.
Lines 228-229: Was there a significant difference in skin temperature and vein diameter between pre and post- HILT values for the clipped group and the non-clipped group? IT would be helpful to have these comparisons.
Lines 243-248: Please include your findings in this paragraph and how this relates to your references.
Line 257: Even if you found vasodilation in both groups, was this a significant change in both?
Line 260: It would be good to comment here if the skin pigment was the same or different between clipped and non-clipped groups.
Line 261: Please include the strength of this correlation. In the results section, the value was not strong.
Lines 276-278: This statement would be stronger if a significant change in pre and post-values for the clipped and unclipped groups.
